# Refining the Clinical Spectrum of the 17p13.3 Microduplication Syndrome: Case-Report of a Familial Small Microduplication

**DOI:** 10.3390/biomedicines10123078

**Published:** 2022-11-30

**Authors:** Jorge Diogo Da Silva, Diana Gonzaga, Ana Barreta, Hildeberto Correia, Ana Maria Fortuna, Ana Rita Soares, Nataliya Tkachenko

**Affiliations:** 1Centro de Genética Médica Doutor Jacinto Magalhães (CGM), Centro Hospitalar Universitário do Porto, 4050-106 Porto, Portugal; 2Life and Health Sciences Research Institute (ICVS), School of Medicine, University of Minho, 4710-057 Braga, Portugal; 3ICVS/3B’s—PT Government Associate Laboratory, 4806-909 Braga, Portugal; 4Centro Materno-Infantil do Norte, Centro Hospital Universitário do Porto, 4099-001 Porto, Portugal; 5Medical Genetics Service, Joaquim Chaves Saúde, 2685-145 Oeiras, Portugal; 6Unit for Multidisciplinary Research in Biomedicine, Abel Salazar Biomedical Sciences Institute, Porto University, 4050-345 Porto, Portugal

**Keywords:** 17p13.3 microduplication, neurodevelopment disorder, intrafamilial variability

## Abstract

The chromosomal region 17p13.3 contains extensive repetitive sequences and is a well-recognized region of genomic instability. The 17p13.3 microduplication syndrome has been associated with a clinical spectrum of moderately non-specific phenotypes, including global developmental delay/intellectual disability, behavioral disorders, autism spectrum disorder and variable dysmorphic features. Depending on the genes involved in the microduplication, it can be categorized in two subtypes with different phenotypes. Here, we report a case of a 7-year-old boy with global developmental delay, speech impairment, hypotonia, behavioral conditions (ADHD and ODD), non-specific dysmorphic features and overgrowth. Genetic testing revealed a small 17p13.3 chromosomal duplication, which included the *BHLHA9*, *CRK* and *YWHAE* genes. Additionally, we observed that this was maternally inherited, and that the mother presented with a milder phenotype including mild learning disabilities, speech impairment and non-specific dysmorphic features, which did not significantly affect her. In conclusion, we present a clinical case of a 17p13.3 duplication that further delineates the clinical spectrum of this syndrome, including its intrafamilial/intergenerational variability.

## 1. Introduction

Chromosomes are highly dynamic structures that frequently recombine with each other and within themselves. While certain chromosomal regions are very stable, others have unusually high rates of recombination: these are a likely consequence of the presence of repetitive DNA sequences in the genome, such as segmental duplications, short interspersed nuclear elements (SINEs) and long interspersed nuclear elements (LINEs) [1]. The presence of these sequences predisposes to non-allelic homologous recombination (NAHR) that can generate deletions and/or duplications of gene-containing regions [1].

The chromosomal region 17p13.3 has a high density of SINEs, specifically of Alu repeats, corresponding to 30% of the DNA sequence (compared to an average of 10% in the whole genome) [2]. This renders this region highly susceptible to recombination and to the generation of Alu-Alu-mediated copy-number variants (CNVs). As the 17p13.3 region is also gene-rich, these CNVs are frequently impactful for the normal function of an organism [3]. In fact, 17p13.3 deletions and/or duplications are frequently reported as pathogenic and are associated with a widespread set of clinical presentations showing variable expressivity [3,4].

Duplications of chromosome 17p13.3 have been classically subdivided into class I and class II, whether duplication of the *PAFAH1B1* gene is absent or present, respectively [5]. Both types of duplications cause developmental delay and behavioral disorders. Specifically, class I microduplications typically show features of autism spectrum disorder (ASD), craniofacial dysmorphisms, hand/foot deformities and growth abnormalities [3,5]. Class II microduplications are less exuberant, causing psychomotor developmental delay and hypotonia [3,5]. Despite this attempt of genotype-phenotype correlation, both types of duplications are extremely variable when it comes to clinical manifestations. This is likely because different duplications affect different sets of genes with different functions, thus varying the presentation of the chromosomal abnormality [6].

Therefore, it is essential to report the clinical spectrum of different 17p13.3 duplications, in a growing effort of pinpointing more precise genotype-phenotype correlations. Here, we describe a patient with a small 17p13.3 duplication that was maternally inherited. The key goal was to further enrich and expand the spectrum of phenotypic characteristics associated with this CNV and show that these duplications also have intrafamilial variable expressivity.

## 2. Case Report

The proband was a 7-year-old Caucasian boy, the only child of non-consanguineous parents. Pregnancy, birth, and the perinatal period were uneventful. During the first years of life he had recurrent respiratory infections, mostly bronchiolitis. He also had chronic nasal obstruction symptoms, requiring tonsillectomy and adenoidectomy with tympanostomy tubes at the age of 4. Additional medical problems also include non-allergic asthma and rhinitis.

He was first evaluated at the Neurodevelopmental Unit at 3 years and 6 months of age. His neurological examination was remarkable by revealing global hypotonia, including orofacial hypotonia with sialorrhea, clumsiness without obvious focal signs, and delay in the acquisition of fine motor skills. He also had speech and language impairment, with normal hearing tests. He showed hyperkinetic behavior and temper tantrums. Psychometric evaluation performed with Griffiths Mental Developmental Scale, at the age of 4 years and 8 months, showed a mild global developmental delay with a General Developmental Quotient (GDQ) of 77. He was referred to speech and language therapy, occupational therapy, and an early stimulation program at preschool.

Growth was normal to the age of 5, but height and weight were above the 99th centile afterwards. At 6 years and 4 months of age, weight was 38 kg (>99th centile, SDS +4.05), height was 130 cm (99th centile, SDS +2.37), body mass index was 22.5 kg/m^2^ (>99th centile, SDS +3.66), and occipitofrontal circumference was 52 cm (48th centile, SDS −0.02), indicating macrosomia and obesity. At 6 years and 8 months he was diagnosed with attention-deficit/hyperactivity disorder (ADHD) and oppositional defiant disorder (ODD).

The physical examination was remarkable for revealing several dysmorphic features (Figure 1A–F): brachycephaly, broad forehead with frontal bossing, hair whorls and hair crest; facial hypotony and sialorrhea; midface hypoplasia with short nose and anteverted nostrils; small ears, small mouth, tented upper lip, downturned corners of the mouth and micrognathia. Additional findings were inverted nipples, long and thin fingers, and fifth finger clinodactyly.

Diagnostic investigation included laboratory analysis for metabolic disorders (ammonium, lactic acid, pyruvate, creatine kinase, isoelectric focusing of serum transferrin, creatine and guanidinoacetate in urine), thyroid function and immunological tests, all of which were normal. The patient had a normal male G-band karyotype and number of CGG repeats in the *FMR1* gene. A comparative genomic hybridization (CGH) array was carried out using an Affymetrix^®^ CytoScan HD array (Santa Clara, CA, USA). The array-CGH detected a 249 kb copy number gain in the 17p13.3 region (arr [hg 19] 17p13.3(1147999_1397019) × 3), which contains five known protein-coding genes (*BHLHA9*, *CRK*, *MYO1C*, *TUSC5* and *YWHAE*) and was considered a variant of likely pathogenic significance (Figure 1G–H). Later array-CGH analyses carried out on the mother determined that this CNV was maternally inherited.

The patient’s mother was a 35-year-old Caucasian female with no previous medical problems. She presented learning disabilities and the need for special education during the school years. Currently, she has mild speech and language impairment, and fine motor skills difficulties, with mild functional impact on her daily activities, such as in her job as a retail worker. Her cognitive function was evaluated by the Wechsler Adult Intelligence Scale, Third Edition (WAIS-III), with a normal result.

Physical examination of the mother was remarkable for dysmorphic features, including telecanthus, midface hypoplasia, small bulbous nose, small ears and clinodactyly. Weight was 63 kg (76th centile, SDS +0.70), height was 161 cm (37th centile, SDS −0.33), body mass index was 24.3 kg/m^2^ (80th centile, SDS +0.83), and occipitofrontal circumference was 54 cm (33rd centile, SDS −0.43). Besides genetic testing, no additional diagnostic tests were performed.

## 3. Discussion

Here we presented two cases of a class I microduplication of the 17p13.3 region in the same family. The proband presented with global developmental delay, speech impairment, hypotonia, behavioral conditions (ADHD and ODD), non-specific dysmorphic features and overgrowth. All these findings have been described in some patients with this type of duplication. Additionally, we observed that this CNV was maternally inherited, and that the mother presented with a milder phenotype which included learning disabilities, speech impairment and non-specific dysmorphic features. The observed dysmorphic features were partially overlapping between the proband and the mother.

Previously described cases of class I 17p13.3 microduplications were shown to mostly occur de novo, whereas a smaller percentage was maternally inherited, as in this case [5]. Currently, there are no literature reports of paternal inheritance. It is interesting to observe that the clinical phenotype of the proband was much more severe than that of the mother, which had only mild learning difficulties during childhood, and normal cognitive function as an adult. Reports of phenotype worsening in the offspring are common for several well-known CNVs, such as 15q13.3 microdeletion [8] and 16p13.11 microdeletion [9], and can reflect ascertainment bias, variable expressivity and/or incomplete penetrance of phenotypes. Nevertheless, the fact that several cases were only diagnosed in adulthood due to an affected descendant indicates that severity might increase over generations, and/or reflect the very mild phenotypes observed in some patients.

A correlation between duplicated genes and the associated phenotype of 17p13.3 microduplication has been proposed for some of the involved genes [3] (Table 1). One of the best-described is *YWHAE* which encodes protein 14-3-3ε that is required for correct migration of neurons in the central nervous system [10]. Duplications of this gene have been associated with ASD and/or developmental delay, the latter being compatible with the findings in our patients, albeit little specific [10].

The gene *CRK* encodes an adaptor protein that is required for cell growth and proliferation: while loss-of-function of this gene is associated with restricted growth, duplications lead to overgrowth/macrosomia, which is observed in the proband but not in his mother [15]. Nonetheless, the proband’s non-affected father had a history of obesity that required bariatric surgery, which could have also genetically contributed to excess weight in his son. Furthermore, duplications of the *BHLHA9* gene are associated with split hand/foot disorders, which were not observed in both our cases [26]. Therefore, this further strengthens the notion that 17q13.3 has a highly variable expressivity, and that other genetic and environmental factors are likely to modulate which characteristics are manifested. No phenotype for duplications of *MYO1C* and *TRARG1* have been yet described.

In conclusion, we presented a case of a small 17p13.3 duplication that mirrors the high variability of symptoms, including in the same family, and that follows a similar trend of other CNVs, with lower severity when they occur de novo. Furthermore, we could observe that, while genotype-phenotype correlations have been partially attempted, it cannot be predicted whether patients will show all symptoms associated with a certain genotype. This adds to the relevance of recognizing all the spectrum of developmental and behavioral phenotypes of 17p13.3 duplications in order to provide guidance for patients’ follow-up and management. Finally, this is also very important when it comes to genetic counselling, as this is currently very hard due to the little understanding of the variability of these types of conditions.

## Figures and Tables

**Figure 1 biomedicines-10-03078-f001:**
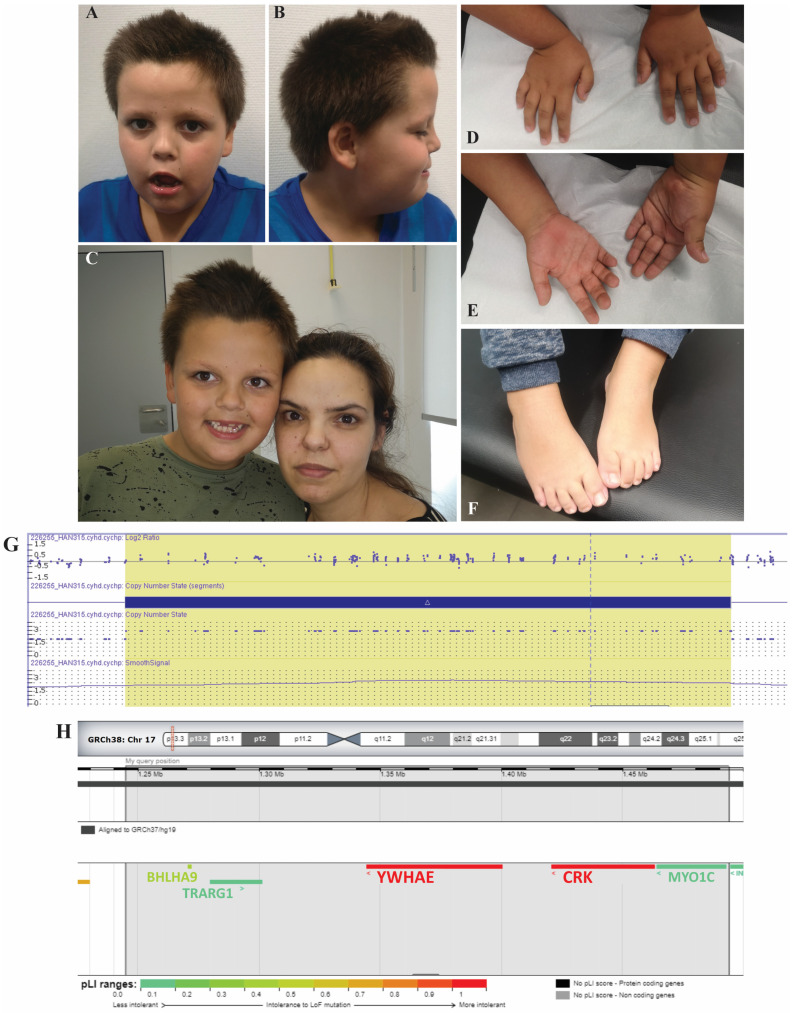
(**A**,**B**) Photographs of the proband to show dysmorphic features. (**C**) Photograph of the proband and its affected mother. (**D**–**F**) Photographs of the hands and feet of the proband. It is noteworthy the absence of any evident split hand/foot malformation. (**G**) Graphical representation of the array−CGH assay of the proband, depicting the duplicated segment, using the Affymetrix^®^ Chromosome Analysis Suite (ChAS) v4.0.0.385 (r28959) with NetAffx 33.2 (hg19). (**H**) Genomic map of the duplicated region and the respective affected genes. The color scale shows the pLI (probability of being loss−of−function intolerant) for each individual gene. Adapted from DECIPHER [7].

**Table 1 biomedicines-10-03078-t001:** Summary of reported phenotypes in human and mice for the affected genes in the clinical case. “?”, unknown or not reported.

Gene	Protein	Gene Function	Gene Duplication	Monoallelic LoF	Biallelic LoF	LoF in Mice
*BHLHA9*	Basic Helix-Loop-Helix Family Member A9	Transcription factor that regulates limb development [11]	Split hand/foot malformation with long-bone deficiency [12]	No phenotype [13] (pLI = 0.30)	Mesoaxial synostotic syndactyly with phalangeal reduction [13]	Syndactyly in the forelimb bud and poliosis in the hindlimb finger [11]
*CRK*	Adapter Molecule Crk	Tyrosine kinase signaling for cell growth and migration [14]	Overgrowth due to growth factor increase [15]	Intrauterine and postnatal growth failure [16] (pLI = 0.96)	?	Embryonically lethal [17]
*MYO1C*	Myosin 1C	Actin-binding motor protein that regulates cell trafficking [18]	?	Sensorineural hearing loss? (unconfirmed) [19] (pLI = 0.00)	?	Loss of retinal photoreceptors [18]
*TRARG1*	Trafficking Regulator of GLUT4	Regulates insulin-stimulated GLUT4 trafficking and insulin sensitivity [20]	?	? (pLI = 0.00)	?	Increased body weight and insulin resistance [21]
*YWHAE*	14-3-3 Protein Epsilon	Adaptor protein for phosphoprotein signaling [22]	Autism spectrum disorder, developmental delay [23]	Brain abnormalities, learning disabilities, and seizures [24] (pLI = 0.98)	?	Cortical and hippocampus malformation [25]

## Data Availability

Not applicable.

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
