# Peer review of "Refining the Clinical Spectrum of the 17p13.3 Microduplication Syndrome: Case-Report of a Familial Small Microduplication"

_biomedicines, 2022, doi:10.3390/biomedicines10123078_

Round 1

Reviewer 1 Report

Paper by Da Silva et al. present a well described case report of hereditary 17p13.3 microduplication. Methodological approach is adequate and discussion of data well supported by references. So I think that paper might be suitable for publication after minor typo error correction (e.g. name of genes should be in Italics).

Author Response

"Paper by Da Silva et al. present a well described case report of hereditary 17p13.3 microduplication. Methodological approach is adequate and discussion of data well supported by references. So I think that paper might be suitable for publication after minor typo error correction (e.g. name of genes should be in Italics)."

We thank the reviewer for his comments and suggestions.

We have now revised all typos in the manuscript (the final version has the respective tracked changes), and have placed all gene names in italics, as suggested.

Thank you for the time taken to assess our work.

Reviewer 2 Report

Da Silver et al here reported two clinical cases of the class I 17p13.3 microduplication. The manuscript is written in a clear and concise manner. It detailed the diagnostic history of the patients and familial manifestation of the syndrome. This work is very relevant to the biomedical field and provides great significance for future genetic counseling. I have a few minor points that the authors should address before acceptance of this manuscript.

1. Since this case report revolves around 17p13.3 microduplication, the authors could consider placing the “Gene Duplication” column right after the “Gene Function” column.

2. Citation 7 in the reference list has a small formatting error.

Author Response

"Da Silver et al here reported two clinical cases of the class I 17p13.3 microduplication. The manuscript is written in a clear and concise manner. It detailed the diagnostic history of the patients and familial manifestation of the syndrome. This work is very relevant to the biomedical field and provides great significance for future genetic counseling. I have a few minor points that the authors should address before acceptance of this manuscript."

We thank the reviewer for the positive comments.

  1. Since this case report revolves around 17p13.3 microduplication, the authors could consider placing the “Gene Duplication” column right after the “Gene Function” column.

As suggested, we have changed the order of the columns in the Table.

  1. Citation 7 in the reference list has a small formatting error.

We have now corrected this error.

Thank you for taking the time to assess and review our work.

Reviewer 3 Report

In this study, Jorge et al, reported a case of 17p13.3 microduplication syndrome with detailed phenotype information. In additiona, the authors further found this genetic variants was maternally inherited, and the mother presented with a milder phenotypes. This study provided important information about the clinical spectrum of 17p13.3 syndrome and its intrafamilial/intergenerational variability. The paper was written in a clear and logical manner, and the analysis and discussion are thorough. However, from the reviewer's personal point of view, there is one major concern need to be addressed before this manuscript can be considered for acceptance. 

1, The publication of patient images should be allowed only if the ethics/privacy consent process has been conducted properly. In the case of patient or relatives have difficulties in following the highly technical language.

Author Response

In this study, Jorge et al, reported a case of 17p13.3 microduplication syndrome with detailed phenotype information. In additiona, the authors further found this genetic variants was maternally inherited, and the mother presented with a milder phenotypes. This study provided important information about the clinical spectrum of 17p13.3 syndrome and its intrafamilial/intergenerational variability. The paper was written in a clear and logical manner, and the analysis and discussion are thorough. However, from the reviewer's personal point of view, there is one major concern need to be addressed before this manuscript can be considered for acceptance. 

We appreciate the positive comments towards our work.

1, The publication of patient images should be allowed only if the ethics/privacy consent process has been conducted properly. In the case of patient or relatives have difficulties in following the highly technical language.

We appreciate the concern of the reviewer on this essential matter. Upon submission of the article, we have also sent the Editor an official consent form from our hospital to attest that the patients allow for publication of their clinical case and photographs (I believe reviewers did not have access to this document). All of the implications of this were explained to the patients in a Medical Genetics consultation, where the technical language was adapted to the cognitive abilities of the patients, as it is required. The informed consent was also signed during this consultation. Therefore, the patients were able to provide consent after being well-informed of all the associated implications.

Thank you for the time taken to assess and review our work.